# Relationship between Sexual Violence and the Health of Spanish Women—A National Population-Based Study

**DOI:** 10.3390/ijerph20043365

**Published:** 2023-02-14

**Authors:** Selene García-Pérez, Guadalupe Pastor-Moreno, Isabel Ruiz-Pérez, Jesús Henares-Montiel

**Affiliations:** 1Unit of Preventive Medicine and Public Health, Puerto Real University Hospital, 11510 Cádiz, Spain; 2Andalusian School of Public Health, Andalusian Government, 18011 Granada, Spain; 3Consorcio de Investigación Biomédica y en Red de Epidemiología y Salud Pública (CIBERESP), 28029 Madrid, Spain; 4Instituto de Investigación Biosanitaria de Granada (ibs. GRANADA), 18012 Granada, Spain

**Keywords:** sexual violence, sexual harassment, cross-sectional survey, health status, suicidal ideation, Spain

## Abstract

Background: Health consequences are likely to be different when sexual violence is analysed independently from other types of violence. It is also likely that different health consequences will result in the cases of partner or ex-partner sexual violence, non-partner sexual violence and sexual harassment. Methods: This study is based on the 2019 Macro-survey of Violence against Women conducted by the Spanish Ministry of Equality on a sample of 9568 women aged 16 years or older. Odds ratios were calculated, and multinomial logistic regression analyses were performed. Results: The present study estimates that 4 out of 10 surveyed women had experienced some form of sexual violence in their lifetime. Sexual harassment is the most frequently reported form of this violence, while intimate partner sexual violence is the form with the most unfavourable sociodemographic characteristics and the worst health impact indicators, such as a greater likelihood of suicidal behaviour. Conclusions: Sexual violence is a widespread, under-studied problem with negative health impacts. Women exposed to intimate partner violence are the most vulnerable and at risk. It is advised that responses and comprehensive care plans be developed that place special emphasis on the protection of victims’ mental health.

## 1. Introduction

The World Health Organization (WHO) defines sexual violence (SV) as “any sexual act, attempt to obtain a sexual act, unwanted sexual comments or advances, or acts to traffic or otherwise directed against a person’s sexuality using coercion, by any person regardless of their relationship to the victim, in any setting, including but not limited to home and work” [1].

The majority of victims of this type of violence are girls and women [2]. As is also the case with all other types of violence against women, sexual violence must also be contextualised within the framework of a patriarchal society, unequal gender structures and sexism [3].

SV against women occurs at all stages of life and is perpetrated by different men (those know to the victim, relatives or strangers). SV perpetrated by different individuals towards a partner or expartner (NPSV) has been examined in different contexts [4,5,6,7,8,9,10], particularly in the case of its most serious manifestations (such as rape) [11,12]. SV of an intimate partner or ex-partner (PSV) is more challenging to study: given that it has traditionally been analysed together with physical violence [13], it often cannot be moulded to fit social definitions of rape, and cultural norms mean women tend not to speak about it [14,15].

Sexual harassment (SH) refers to any type of undesired sexual behaviour or the solicitation of sexual favours, including verbal and physical behaviours, or any type of gesture of a sexual nature [16]. SH has been studied in different ambits, including in educational [17,18] and work settings [19,20] and, more recently, online [21,22]. Further, interest in this topic has increased in recent years due to the media and social impact of movements such as #MeToo [23,24].

Estimates from data gathered using different populational surveys show that the prevalence of PSV throughout life varies from 3.3% to 18.3%, whilst the prevalence of NPSV ranges from 6.2% to 52.2% [25,26,27]. With regard to the prevalence of SH, studies demonstrate that a significant number of women are sexually harassed at some point during their life, with a prevalence of 81% in the USA [28] and 84% in the United Kingdom [29].

Variation seen in the frequencies reported by different studies may be explained by the diverse definitions of SV used in the different samples examined [30]. In many cases, definitions of SV necessarily refer to subjective experiences [31]. This might mean that outcomes are not always comparable, given that meaningful differences may exist as a function of the perception held by a population when surveyed regarding sexual violence and harassment and its limits [32].

When SV is analysed separately from other types of violence (such as physical or psychological violence), social determinants are uncovered that differ from those identified in the existing literature on gender violence overall [33]. For this reason, it is appropriate to distinguish between the different forms of SV (PSV, NPSV and SH) and examine the social determinants related to each one of these forms. This is necessary to identify the underlying social mechanisms and, in this way, be able to develop prevention strategies focused on their origin. If such work is not conducted, it is likely that the prevalence of SV will be underestimated [34].

Further, whilst it is true that physical violence can have a more immediate and visible impact (such as bruises, breaks, cuts, etc.), research demonstrates that violence that is psychological and sexual in nature is also independently related to health issues, such as post-traumatic stress disorder, depression, psychosis and substance abuse problems [35,36,37].

Thus, health consequences are likely to be different when SV is analysed independently from other types of violence. It is also likely that different health consequences will result in the cases of PSV, NPSV and SH given that the dynamics, intensity and longevity pertaining to each one will likely be different [34].

The majority of published studies on this issue use different samples to examine the consequences of each of the different forms of SV on health, given that it is not easy to find sources that provide such information broken down within a single given population. In this sense, the Macro-survey of Violence against Women in Spain provides a huge opportunity. This survey was conducted by the Spanish Government and provides estimates of different types of gender violence in a sample of women older than 16 years [38].

The 2019 Macro-survey incorporates novel aspects with regard to prior surveys. For example, it increases the number of variables included in the unit on NPSV and introduces a new unit on SH. As a result, at the time of writing, it is the only official statistical setup that allows us to make estimations about the different types of SV in a single sample of women in Spain. 

The present work aims to identify the health impact on victims of SV in Spain using the latest data available from the Macro-survey of Violence against Women. To this end, relationships were examined between three types of SV (PSV, NPSV and SH) and five health indicators: self-perceived poor health, consumption of psychopharmaceuticals, suicidal ideation and use of health services.

## 2. Materials and Methods

The Macro-survey of Violence against Women refers to a study conducted by the Spanish Ministry of Equality and Sociological Research Center on a sample of 9568 women over the age of 16 years who reside in Spain [38]. This survey used multistage stratified cluster sampling, with proportionate random selection of the primary (municipalities) and secondary sampling units (sections) and random-route selection of the final units (individuals) with age and occupation quotas.

Data were collected between September and December 2019 from computer-assisted personal interviews (CAPI) conducted in participating women’s homes. Response cards were used for the most sensitive questions to facilitate privacy. All women participated voluntarily and did not receive any reward or reimbursement. All data on examined variables and frequency distributions recorded by the macro-survey can be consulted at the following website: http://www.cis.es/cis/export/sites/default/-Archivos/Marginales/3220_3239/3235/es3235mar.html (accessed on 21 October 2022).

For statistical treatment, three variables were defined with respect to self-reported sexual violence:(a)Partner or ex-partner sexual violence (PSV).(b)Non-partner sexual violence (NPSV).

PSV and NPSV were measured using eight items (Appendix A). Women responding affirmatively to one of the eight items were considered to have experienced PSV or NPSV.

(c)Sexual harassment (SH).

SH was measured using eleven items (Appendix A). Women who had experienced this form of violence were identified in accordance with affirmative responses that they had experienced unwanted behaviours with a sexual connotation from any other individual. Women responding affirmatively to any one of these items were considered to have experienced SH.

Given that these three variables are not mutually exclusive, participating women had to indicate whether they had experienced situations pertaining to all three over the course of their lifetime.

In line with previously conducted scientific research, the following sociodemographic variables were included [1,39,40]: age (16–25/26–45/46–65/≥66); size of municipality (≤10,000/10,001–100,000/≥100,001 inhabitants); highest educational attainment (primary or less/secondary (high school)/higher (university)); parental status (has children/does not have children); country of birth (Spain/other); employment status (unemployed or not employed outside of the home/employed/retired/student); level of family income (≤EUR 300 per month/EUR 301–900 per month/EUR 901–1800 per month/EUR 1801–3000 per month/>EUR 3000 per month); and having a confidant (social support) (no/yes).

With regard to health outcomes, we selected a group of variables that were available for the three forms of violence analysed. The following variables were included: suicidal ideation throughout life (yes/no); suicide attempt throughout life (only asked if the women answered yes to the question on suicidal ideation) (yes/no); consumption of antidepressives or tranquilisers in the last 12 months (yes/no); self-perceived health state in the last 12 months (good/bad); use of health services in the last 12 months; use of emergency services (yes/no); hospital stay (yes/no); and visit to a psychologist/psychotherapist/psychiatrist (yes/no).

Prevalence of PSV, NPSV and SH was calculated over the course of the lifetime. For bivariate logistic analysis, the chi-square test was used to analyse the relationships between categorical variables. Adjusted ORs were calculated, and binomial logistic regression analyses were performed in order to identify the types of violence that were associated with each health outcome. Statistical significance was set at *p* < 0.05. All analyses were performed using R statistical software. Weightings included in the 2019 Macro-survey of Violence against Women were used for analysis.

## 3. Results

Sociodemographic variables pertaining to the study sample are presented in Table 1. The average age of participating women was 49.9 years old (SD: 18.3 years)—the youngest participant was 16 years old, and the oldest was 96 years old. The age group with the most participants was that made up of those aged 46–65 years (33.8% of the sample). With regard to educational level, 52.4% had finished secondary school, with fewer women reporting their highest educational attainment to be primary school or having no formal education (24.8%) and even fewer reporting having a higher education (22.8%). The most common occupational status reported was employed (44.2%), followed by unemployed or not employed outside of the home (27.8%). The most commonly reported average income was EUR 901–1800/month (37.7%), followed by EUR 1801–3000/month (28.8%). A total of 43.1% lived in municipalities with 10,001–100,000 inhabitants. The majority were of Spanish nationality (91.7%), had children (72.2%) and mentioned that they had a close confidant with whom they could talk in private (94.0%).

### 3.1. Frequency of the Three Forms of Sexual Violence throughout Life

Of the 9568 women interviewed, 4233 (44.24%) had been victims of one or more of the different forms of SV examined in the present study. In total, 40.37% had suffered SH, 8.86% had suffered PSV (current or ex-partner/s) and 6.47% had suffered NPSV at some point during their lifetime.

Figure 1 shows the number of women who had suffered the different types of SV. The frequency of SH alone was recorded to be 30.52%, whilst PSV and NPSV alone in the present sample were 2.96% and 0.76%, respectively. Fourteen women (0.15%) reported having suffered PSV and NPSV; 411 women (4.30%) had suffered PSV and SH; and 392 (4.10%) had suffered NPSV and SH. Finally, 140 women (1.46%) had been victims of all three examined types of SV at some point during their lives. 

### 3.2. Sociodemographic Characteristics of Women Who Have Suffered Sexual Violence

Sociodemographic characteristics of participating women who had suffered SV are presented in Table 2.

With regard to the average age of the women, similar outcomes were found for those who were victims of SH and NPSV, being 43.8 years old (SD: 16.9) and 42.9 years old (SD: 15.8), respectively; however, victims of PSV were typically older, with an average age of 46.7 years old (SD: 16.8). This group was also made up of a greater percentage of women older than 66 (15.5%) and contained the lowest percentage of young women (12.1%).

Regarding educational level, secondary education was found to be the most common highest educational attainment for women for all three types of SV. However, whilst the second most commonly reported highest educational attainment for women who were victims of SH and NPSV was higher or university education (30.9% and 33.9%, respectively), this place was occupied by primary education for women who had suffered PSV (22.8%).

Concerning financial income, the groups corresponding to SH and NPSV reported similar outcomes, with the most commonly reported average monthly income being EUR 901–1800, followed by EUR 1801–3000 and, thirdly, >EUR 3000. Nonetheless, the means reported in the group of women who were victims of PSV were lower, with EUR 901–1800/month being most often reported, followed by EUR 301–900/month. In the final group, victims of PSV, it is notable that the proportion of women earning more than EUR 3000/month was only 8.4%, compared to 16.8% and 17.2% found in relation to the other types of SV.

With regard to job level, in the case of the three forms of SV, women tended to mainly be employed, followed by unemployed or not employed outside of the home. In the group of victims of PSV, the lowest proportion of students was observed, with 5.1% relative to 10.2% of victims of SH and 8.3% of victims of NPSV.

With regard to parental status, the most common finding for all three types of SV was that victims had children, with the group of victims of PSV standing out as having the highest percentage, specifically 75.3% relative to 61.6% of women who had suffered SH and 59.4% of those who had suffered NPSV.

With regard to place of residence, the majority of victims in the three SV groups lived in localities that were of an intermediate size (10,001–100,000/inhabitants), with a slightly higher percentage of very small localities (≤10,000/inhabitants) pertaining to PSV, specifically 19.2% relative to 17.5% seen in SH and 16.8% in NPSV groups.

Concerning the country of birth, a higher percentage of victims of PSV were observed to have been born in countries outside of Spain, with 11.4% relative to 8.7% of SH victims and 9.7% of NPSV victims.

Finally, with regard to the social support variable, the majority of victims of all three types of SV reported having a person close to them whom they could confide in, with the percentage of “yes” responses standing out as being slightly higher in the case of SH than the other groups (94.5%).

### 3.3. Health Impact of the Three Types of Sexual Violence

Examination of the relationship between the three types of sexual violence and indicators of their health impact produced the outcomes presented in Table 3.

Of those women who had experienced SH, 28.4% perceived themselves to be in poor health (aOR = 1.36, 95%CI [1.19–1.55]); 19.9% reported having taken psychopharmaceuticals in the last 12 months (aOR = 1.39, 95%CI [1.20–1.60]); and 13.5% had thought about suicide on at least one occasion (aOR = 2.14, 95%CI [1.75–2.61]), with 28.1% of those reporting they had attempted suicide (aOR = 2.60, 95%CI [1.79–3.81]).

In the case of women who reported NPSV, 36.2% perceived themselves to be in poor health (aOR = 1.62, 95%CI [1.28–2.04]); 27% reported having taken psychopharmaceuticals in the last 12 months (aOR = 1.66, 95%CI [1.30–2.11]); and 28.1% had thought about suicide on at least one occasion (aOR = 2.75, 95%CI [2.12–3.56]), with 34.8% of those reporting they had attempted suicide (aOR = 2.05, 95%CI [1.31–3.14]).

In addition, 41.9% of women who reported PSV perceived their health to be poor (aOR = 1.66, 95%CI [1.37–2.02]); 32.7% reported having taken psychopharmaceuticals in the last 12 months (aOR = 1.93, 95%CI [1.58–2.34]); and 27.9% had thought about suicide on at least one occasion (aOR = 3.00, 95%CI [2.40–3.74]), whilst 37% of those mentioned had attempted it (aOR = 4.25, 95%CI [2.96–6.08]).

Table 4 examines the relationship between sexual violence and the use of health services in the last 12 months. 

Women who had experienced SH more frequently reported having made use of emergency services (aOR = 1.56, 95%CI [1.38–1.76]) and a psychologist, psychotherapist or psychiatrist (aOR = 1.75, 95%CI [1.46–2.09]) than those who had not experienced this type of violence.

Women who stated that they had suffered NPSV also more frequently reported having made use of emergency services (aOR = 1.68, 95%CI [1.36–2.07]) and having visited a psychologist, psychotherapist or psychiatrist (aOR = 1.92, 95%CI [1.48–2.47]) than those who did not report having experienced this type of violence.

In the case of women who had experienced PSV, in addition to more frequently reporting having attended emergency services (aOR = 1.49, 95%CI [1.24–1.78]) and seen a psychologist, psychotherapist or psychiatrist (aOR = 1.87, 95%CI [1.48–2.33]), they also more often reported having had a hospital stay (aOR = 1.79, 95%CI [1.41–2.27]) than those who had not experienced PSV.

## 4. Discussion

The present work highlights the frequency of three types of SV in a representative sample of women in Spain and the impact of this on its victims. Almost 45% of the women surveyed had suffered some type of SV throughout their lifetime. When analysing the different types of SV separately, SH was reported by 40% of the surveyed women, whilst 8.9% reported PSV and 6.5% reported NPSV.

Previously conducted research focusing on combined measures of partner violence has reported that younger age, unemployment, lower educational or socioeconomic status, having a greater number of children, being an immigrant and a lack of social support all increased the likelihood of violence against women [40,41,42]. Nevertheless, analysis of the different forms of SV gathered in the Macro-survey of Violence against Women revealed other social determinants.

Thus, the present work identified the sociodemographic characteristics of women exposed to PSV: they tend to be older, have children, be women who were born outside of Spain and have a lower educational level and socioeconomic status compared to those exposed to NPSV and SH. These sociodemographic characteristics identify particularly vulnerable groups. From an intersectional perspective, this should make us reflect on the disadvantages or obstacles that these women may encounter from an integral perspective, and avoid simplifying the conclusions and, therefore, the approach to this reality. 

Present findings reported here are in line with those reported by the few previously conducted studies based on populational surveys that examined the characteristics of women who suffered the three forms of SV examined in the present work. In Australia, women reporting SH tended to be aged between 18 and 24 years, were not married, had few school qualifications and had financial problems [43]. Data from the USA on NPSV also indicate that women who suffer SH are younger than victims of PSV [44].

The present work also suggests that SV is an important risk factor for health, regardless of the form it takes, highlighting that the victim–aggressor relationship is crucial at the time of considering the impact of SV. 

The group of women who had suffered NPSV were more likely to have attended emergency services and/or seen a psychotherapist. With regard to SH, findings also reveal that exposed women experience negative health repercussions, but despite this typically being a commonly reported antecedent, it appears to have less impact than in the case of the other forms studied. Further, findings indicate that PSV can be especially traumatic, with victims being more likely than those of other forms of SV to have poor self-perceived health, consume psychopharmaceuticals, be taken into hospital, present with suicidal ideation or make a suicide attempt.

Previously conducted studies report analogous findings. For example, Baker [45] highlights that when compared with other forms of interpersonal violence, exposure to PSV is associated with a greater risk of post-traumatic stress syndrome (PTSD) and depressive symptoms, consumption of harmful substances, suicide, pain and other somatic symptoms. Temple [46] stated that sexual aggression perpetrated by a current partner was the strongest predictor of PTSD, stress and disassociation, when compared with sexual aggression perpetrated by a prior partner or other men. SH and its relationship with health is currently the focus of more attention, with studies reporting similar outcomes to those reported in the present work, although research has generally been conducted with specific samples [47,48,49] and in determined settings [50,51].

Further, the presence of PSV, together with other types of partner violence, has been associated with more serious depressive symptoms and a higher incidence of suicide attempts [52,53]. All of this, together with their unfavourable sociodemographic situation, would justify identifying these women as a particularly vulnerable group who suffer from a form of violence that will probably be continuous and prolonged. In such cases, the lack of financial stability, a social network, or information and knowledge about SV play an important role as factors that lead these women to not report their experiences or end the relationship [33,40].

The present findings also highlight the association of suicidal ideation and the act itself with prior exposure to SV. A total of 28% of women who suffered SH reported attempting suicide, with 34.8% of women who suffered NPSV and 37% of those who suffered PSV reporting this.

Suicide is an important health issue; despite this, it often goes ignored. The isolation, submission and silence in which victims of SV live, together with the stigma, ignorance and refusal to talk about suicide, make the existing relationship between both phenomena a reality that goes largely ignored. Tackling the complexity of suicidal behaviour starts with identifying both risk and protective factors [54], and given present findings, women who have suffered SV should be included in the early detection and intervention plans developed by the pertinent public bodies to tackle suicide [55].

It is important to highlight the overlap between the three types of SV analysed, revealing that a meaningful number of women suffer double or even triple victimisation. The health impact on these women will presumably be far more serious, and they will also need greater care to be taken with their personal wellbeing and mental health. This will require more attention both in the literature and on behalf of the care services available to victims. Moreover, the low frequency of PSV (2.96%) could be explained by the normalisation/naturalisation of sexual violence perpetrated by intimate partners, especially within sexist cultures such as southern European ones, which would indicate that the real numbers are much higher and the magnitude of the problem even greater.

As a possible limitation of this study, it should be noted that we cannot rule out the possibility that there are other factors that influence the health status of the women interviewed. Only a certain number of the possible independent variables related to the health effect we wished to study were collected in the survey. In this case, we have adjusted for the most relevant sociodemographic factors and included a social support variable to minimize this confusion as much as possible. Another possible limitation is that the definition of NPSV in the macro-survey is: “sexual violence against women originated by other persons with whom the woman interviewed does not maintain or has not maintained a partner relationship”. The time window used for this question is throughout life, so it cannot be ruled out that violence that occurred in childhood is included in this total, but it is not possible to differentiate between them using the information collected in the survey. Present findings highlight that tackling SV must not lead to its trivialisation in the mass media, with it also being necessary to highlight and explain to society the health impact it has on its victims. 

In Spain, the next Macro-survey of Violence against Women is scheduled to be conducted in 2023 as it is programmed every four years. Given the devastating effects of crisis events and emergency situations (such as the COVID-19 pandemic) on victims of violence, this next macro-survey will be able to show the impact of the pandemic on gender violence in Spain and, presumably, reveal that current trends point to a prevalence that is on the rise [56].

## 5. Conclusions

It can be concluded that SV, in any of its forms, is a serious global public health problem with dangerous side-effects for victims, such as suicidal behaviours, of which the women recounting such actions are survivors. Recent movements on a global scale to overwhelmingly and totally denounce SV have given rise to new expectations which urge greater responsibility and a call for action. Many countries, including Spain, have added explicit consent to their legislative frameworks as a key aspect when judging sexual crimes, with this representing a meaningful step forward. Nonetheless, it is also recommended to continue to investigate the different forms of SV as independent entities (separate from physical violence) and develop appropriate responses to meet women’s needs. For this, it will be necessary to establish immediate basic care plans which place particular emphasis on the protection of mental health. Likewise, it will be crucial for the agents involved in such care, such as social-health professionals, to be aware of and equipped to deal with SV. 

## Figures and Tables

**Figure 1 ijerph-20-03365-f001:**
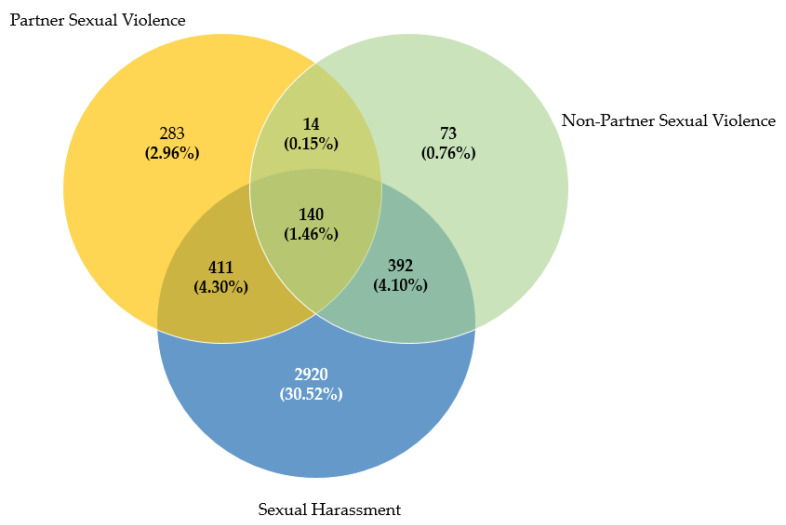
Frequency and relationships between the three forms of sexual violence.

**Table 1 ijerph-20-03365-t001:** Sociodemographic characteristics of the sample (*n* = 9568).

	*n* (%)
Age (years)	
Mean (SD)	(minimum–maximum)
49.96 (18.35)	(16–96)
Age (years)	
16–25	1047 (10.9%)
26–45	3096 (32.4%)
46–65	3233 (33.8%)
≥66	2192 (22.9%)
Educational level	
Primary or without formal education	2366 (24.8%)
Secondary (high school)	5006 (52.4%)
Higher or university	2178 (22.8%)
Job status	
Unemployed/not employed outside of the home	2645 (27.8%)
Student	579 (6.1%)
Employed	4206 (44.2%)
Retired	2088 (21.9%)
Financial income (EUR/month)	
≤300	179 (2.9%)
301–900	1176 (19.0%)
901–1800	2337 (37.7%)
1801–3000	1784 (28.8%)
>3000	719 (11.6%)
Size of municipality (*n* inhabitants)	
≤10,000	2007 (21.0%)
10,001–100,000	4122 (43.1%)
≥100,001	3439 (35.9%)
Country of birth	
Spain	8779 (91.7%)
Other	789 (8.3%)
Children	
Yes	6907 (72.2%)
No	2659 (27.8%)
Social support	
Yes	8975 (94.0%)
No	576 (6.0%)

**Table 2 ijerph-20-03365-t002:** Sociodemographic characteristics of women who have suffered different forms of sexual violence.

	SH	*p* Value	NPSV	*p* Value	PSV	*p* Value
Age (years)						
Mean (SD)	43.8 (16.9)		42.9 (15.8)		46.7 (16.8)	
Age (years)		<0.01 *		<0.01 *		<0.01 *
16–25	630 (17.1%)		101 (17.8%)		96 (12.1%)	
26–45	1439 (39.1%)		227 (40.0%)		303 (38.1%)	
46–65	1171 (31.9%)		186 (32.8%)		274 (34.4%)	
≥66	436 (11.9%)		53 (9.3%)		123 (15.5%)	
Educational level		<0.01 *		<0.01 *		<0.01 *
Primary/without studies	442 (12.0%)		58 (10.2%)		181 (22.8%)	
Secondary (high school)	2094 (57.1%)		316 (55.8%)		458 (57.8%)	
Higher/university	1133 (30.9%)		192 (33.9%)		154 (19.4%)	
Financial income (EUR/month)		<0.01 *		<0.01 *		<0.01 *
≤300	65 (2.7%)		13 (3.3%)		18 (3.2%)	
301–900	313 (13.2%)		65 (16.4%)		152 (26.7%)	
901–1800	831 (35.0%)		122 (30.8%)		223 (39.1%)	
1801–3000	765 (32.2%)		128 (32.3%)		129 (22.6%)	
>3000	400 (16.8%)		68 (17.2%)		48 (8.4%)	
Occupational status		<0.01 *		<0.01 *		<0.01 *
Unemployed/not employed outside of the home	896 (24.5%)		153 (27.1%)		212 (26.8%)	
Employed	1862 (50.9%)		293 (52.0%)		402 (50.8%)	
Retired	524 (14.3%)		71 (12.6%)		138 (17.4%)	
Student	374 (10.2%)		47 (8.3%)		40 (5.1%)	
Parental status		<0.01 *		<0.01 *		<0.05 *
No children	1411 (38.4%)		230 (40.6%)		197 (24.7%)	
Children	2264 (61.6%)		337 (59.4%)		599 (75.3%)	
Municipality (n inhabitants)		<0.01 *		<0.05 *		>0.05 *
≤10,000	642 (17.5%)		95 (16.8%)		153 (19.2%)	
10,001–100,000	1591 (43.3%)		241 (42.5%)		345 (43.3%)	
≥100,001	1443 (39.3%)		231 (40.7%)		298 (37.4%)	
Country of birth		>0.05 *		>0.05 *		<0.01 *
Spain	3358 (91.3%)		512 (90.3%)		705 (88.6%)	
Other	318 (8.7%)		55 (9.7%)		91 (11.4%)	
Social support		>0.05 *		<0.01 *		<0.01 *
No	202 (5.5%)		52 (9.2%)		76 (9.6%)	
Yes	3466 (94.5%)		512 (90.8%)		716 (90.4%)	

* Chi-square test; SH—sexual harassment; NPSV—non-partner sexual violence; PSV—partner or ex-partner sexual violence.

**Table 3 ijerph-20-03365-t003:** Relationship between the three types of sexual violence and indicators of their health impact.

	Poor Self-Perceived Health		Psychopharmaceuticals Consumption		Presence of Suicidal Ideation		Suicide Attempt *	
	*n* (%)	OR_a_ (95%CI)	*n* (%)	OR_a_ (95%CI)	*n* (%)	OR_a_ (95%CI)	*n* (%)	OR_a_ (95%CI)
SH								
No	1919 (32.6%)	1	1101 (18.7%)	1	374 (6.4%)	1	75 (20%)	1
Yes	1045 (28.4%)	1.36 (1.19–1.55)	732 (19.9%)	1.39 (1.20–1.60)	494 (13.5%)	2.14 (1.75–2.61)	139 (28.1%)	2.60 (1.79–3.81)
NPSV								
No	2759 (30.7%)	1	1680 (18.7%)	1	710 (7.9%)	1	159 (22.4%)	1
Yes	205 (36.2%)	1.62 (1.28–2.04)	153 (27%)	1.66 (1.30–2.11)	158 (28.1%)	2.75 (2.12–3.56)	55 (34.8%)	2.05 (1.31–3.14)
PSV								
No	2631 (30%)	1	1573 (17.9%)	1	649 (7.4%)	1	133 (20.5%)	1
Yes	333 (41.9%)	1.66 (1.37–2.02)	260 (32.7%)	1.93 (1.58–2.34)	219 (27.9%)	3.00 (2.40–3.74)	81 (37%)	4.25 (2.96–6.08)

* The percentages of suicide attempt are calculated from the total number of women who answered yes to the presence of suicidal ideation. SH—sexual harassment; NPSV—non-partner sexual violence; PSV—partner or ex-partner sexual violence.

**Table 4 ijerph-20-03365-t004:** Relationship between the types of sexual violence and indicators of their health impact with the use of health services.

	Attend Emergency Services		Psychotherapist Visit		Hospital Stay	
	*n* (%)	OR_a_ (95%CI)	*n* (%)	OR_a_ (95%CI)	*n* (%)	OR_a_ (95%CI)
SH						
No	1506 (25.6%)	1	452 (7.7%)	1	584 (9.9%)	1
Yes	1338 (36.4%)	1.56 (1.38–1.76)	518 (14.1%)	1.75 (1.46–2.09)	401 (10.9%)	1.18 (0.98–1.42)
NPSV						
No	2589 (28.8%)	1	841(9.3%)	1	903 (10%)	1
Yes	255 (45%)	1.68 (1.36–2.07)	129 (22.7%)	1.92 (1.48–2.47)	82 (14.5%)	1.33 (0.98–1.79)
PSV						
No	2512 (28.6%)	1	790 (9%)	1	859 (9.8%)	1
Yes	332 (41.7%)	1.49 (1.24–1.78)	180 (22.6%)	1.87 (1.48–2.33)	126 (15.8%)	1.79 (1.41–2.27)

SH—sexual harassment; NPSV—non-partner sexual violence; PSV—partner or ex-partner sexual violence.

## Data Availability

The data are available upon request from the corresponding author.

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
