# Peer review of "Relationship between Sexual Violence and the Health of Spanish Women—A National Population-Based Study"

_ijerph, 2023, doi:10.3390/ijerph20043365_

Round 1

Reviewer 1 Report

Relationship between sexual violence and the health of Spanish women. A national population-based study

This study is an important examination of sexual violence and health in a large sample of Spanish women. It is based on a national population-based survey and includes a diverse and large sample of women residing in the country. The study provides good data on four types of sexual violence among Spanish women revealing high levels of sexual harassment (40.37%) and lower but significant levels of partner (8.86%) and non-partner (6.47%) sexual violence over lifetime. This supports some research findings from other countries.

I think the study provides new information about sexual violence in Spain and is an important contribution to the area of gender-based violence. The following comments to the authors are suggestions for improved clarity and acknowledgment of study limitations.

The authors state that is it important to examine sexual violence separately from other forms of violence. While this is valid and a good approach, it is not clear from the study that the participants did not also experience physical or psychological and other forms of abuse concurrently with sexual violence. Can this be clarified? This would have implications for determining health effects. Are the health effects associated only with sexual violence? Since we do not know about other aspects of the participants’ lives, it is not possible to conclude that the health effects found in this study are solely attributed to sexual violence. This should be discussed as a limitation to the study. This is related also to the general limitation of the study and other cross-sectional studies that examine variables that are associated with particular events, in this case sexual violence, but cannot conclude that there are causal relationships between sexual violence and identified outcomes. 

The study presents a number of health outcomes or co-occurring health problems. While these are important, they are some limits in terms of understanding the effects of sexual violence that should be addressed by the authors. For example, posttraumatic effects are not included nor are other commonly associated psychological effects such as depression and stress reactions. Suicidal ideation and suicide attempts appear to be lifetime prevalence although this is not specifically indicated. Can the authors clarify this? Other health outcomes refer to the past 12 months only and there is a mix of reporting lifetime and past 12 months among the health services variables as well. Can this be clarified? What can the authors say about lifetime occurrence (e.g., if suicidal ideation occurred during adolescence) of any of the health outcomes and sexual violence? It is not clear what “poor self-perceived health” is. It is noted that women respondents who did not experience sexual violence also reported quite high levels of this self-perception. Can the authors comment on this?

It would be helpful to have more definition and clarity around what the authors mean by NPSV – especially if it is a lifetime estimate. Does this include childhood sexual abuse? Studies often distinguish between adult and child sexual violence so the ages that are included in this study should be identified.

With respect to the job status of respondents, the identity of ‘housewife” should not be included in the unemployed category. The authors can attempt to disentangle this grouped variable and label ‘housewife” in more appropriate terms such as ‘not employed outside of the home”.

Reviewer 2 Report

The study has an interesting sample and it allows to problematize the extend of sexual violence. Although it is addressed as gender violence, in the discussion it is presented as a niche problem which can promote stigmatization [Thus, the present work identified the sociodemographic characteristics of women 280 exposed to PSV are less favorable than those subjected to other types of SV, given that 281 they tend to be older, have children, be women who were born outside of Spain and have 282 a lower educational level and socioeconomic status]. Instead, we strongly suggest that some sociodemographic characteristics can be pointed out as particularly vulnerable groups (e.g., intersectional approach), regarding socially and culturally less access from these women to resources (e.g., migrant women, racialized women, trans women), but it should be unequivocal that being sexual violence a gendered type of violence, potentially harms all women, directly or indirectly, avoiding reductive data readings.

Therefore, we also suggest to underline the normalization/naturalization of sexual violence perpetrated by intimate partners and be addressed as a possible cause to a such a lower percentage of PSV (2.96%), specially within sexist cultures as southern Europe ones.

It should be presented the authors decision on the health impact criteria - why and what underpins each one, specially the “Poor self-perceived health” (physical and mental?) and drug consumption (legal, ilegal? which legal?). Also, it should be presented a time frame regarding the use of health services, otherwise it can be extrapolated an association that might not exist. Was the questionnaire built by the authors or adapted? Did it got throught an ethical committe? 

We also suggest a deepen and critical discussion - e.g., why is the percentage of suicide attempt higher than the suicide ideation?

Round 2

Reviewer 2 Report

It is said that PSV and NPSV were measured using eight items. Women responding affirmatively 117 to one of the 8 items were considered to have experienced PSV or NPSV. It is suggested thar these items are presented? Why is considered VS answering positive to 1/8 of the items? Was the questionnaire developed by the research team? It is necessary to understand the criteria that underline the categorization of SV and the same with the 11 items for SH.

It would also be important to understand the researchers decision on focus the following outcomes and not others - suicidal ideation throughout life; suicide attempt throughout life; consumption of anti-depressives or tranquilizers in the last 12 months; self-perceived health state in the last 12 months; use of health services in the last 12 months; use of emergency services; hospital stay visit to a psychologist/psychotherapist/psychiatrist. Which literature underlined this decision?

Why it is said that women birthed outside of Spain and lower education show higher percentage of SV?

Best Regards,
